# Impact of covid-19 on long-term oxygen therapy 2020: A nationwide study in Sweden

**Josefin Sundh**[1]\*, **Andreas Palm**[2,3], **Josefin Wahlberg**[4], **Michael Runold**[5], **Magnus Ekström**[6]

**1** Department of Respiratory Medicine, Faculty of Medicine and Health, Örebro University, Örebro, Sweden, **2** Department of Medical Sciences, Respiratory, Allergy and Sleep Research, Uppsala University, Uppsala, Sweden, **3** Centre for Research and Development, Uppsala University/Region of Gävleborg, Gävle Hospital, Gävle, Sweden, **4** Department of Medicine, Blekinge Hospital, Karlskrona, Sweden, **5** Department of Respiratory Medicine and Allergology, Faculty of Medicine, Karolinska University Hospital, Stockholm, Sweden, **6** Department of Clinical Sciences Lund, Respiratory Medicine and Allergology and Palliative Medicine, Faculty of Medicine, Lund University, Lund, Sweden

\* josefin.sundh@regionorebrolan.se

**Data Availability Statement:** Data cannot be made freely available as they are subject to secrecy in accordance with the Swedish Public Access to Information and Secrecy Act, but can be made available to researchers upon request, after

## Abstract

### Introduction

Covid-19 can cause chronic hypoxic respiratory failure, but the impact on the need for long-term oxygen therapy (LTOT) is unknown. The aim was to investigate change in incidence and characteristics of patients starting LTOT in Sweden 2020 after the outbreak of the pandemic.

### Material and methods

Population-based observational study using data from the National Registry for Respiratory Failure (Swedevox) and from a survey to all centres prescribing LTOT in Sweden. Swedevox data provided information on incidence of LTOT and characteristics of patients starting LTOT during 2015–2020.

### Results

Between March-Dec 2020, 131 patients started LTOT due to covid-19, corresponding to 20.5% of incident LTOT in Sweden. Compared with 2015–19, the total number of patients starting LTOT did not increase. No significant differences in patient characteristics or underlying causes of hypoxemia were found between patients starting LTOT during 2020 compared 2015–2019. The majority of the LTOT centres estimated that, since the start of the pandemic, the incidence of LTOT was unchanged and the time devoted for LTOT work was the same or slightly less.

### Conclusions

Covid-19 caused one fifth of all LTOT starts during the pandemic in 2020. The LTOT incidence overall did not increase possibly due to reduction in other infections.

approval from the Swedish Ethical Review
Authority has been obtained (https://
etikprovningsmyndigheten.se;
registrator@etikprovning.se).

**Funding:** Magnus Ekström was supported by
unrestricted grants from the Swedish Society for
Medical Research and the Swedish Research
Council (Dnr 2019-02081). There was no additional
external funding received for this study.

**Competing interests:** The authors have no
conflicts of interest related to the study.

## Introduction

Infection with severe acute respiratory syndrome coronavirus 2 (SARS-CoV-2), or covid-19, was declared as a pandemic by WHO in March 2020. By February 2022, the global number of people diagnosed with covid-19 is over 416 million, and almost 5,9 million have died due to the disease [1]. In Sweden, the first covid-19 case was confirmed on the 31st of January 2020, but the disease was not generally spread until March 2020 [2]. At the close of 2020, 462,661 cases of covid-19 had been confirmed in Sweden, and in the end of May 2021 the cumulative incidence had increased to 1 068,473 cases [2] (Fig 1). The underlying population in Sweden is 10,23 million people, and subsequently the proportion of the population diagnosed with covid-19 was 4.5% at the end of 2020 and 10,4% in May 2021.

In patients hospitalized for covid-19, mortality in hospital varies from 22% to 38% [3–6]. Survivors from severe covid-19 infection may, through damage on the lung parenchyma or pulmonary vasculature, develop chronic hypoxic respiratory failure [7]. Among patients hospitalized due to a covid-19 infection, the proportion with a remaining impaired diffusion capacity of the lung for carbon monoxide (DLCO) below 80% has been reported to be 52% after four months [8, 9] and 29% after six months [10]. In a longer perspective, the most important consequence is development of chronic respiratory failure in need of continued oxygen supply. Several studies have reported both diminished radiological findings and increased DLCO over time after a COVID-19 pneumonia, but the minority of patients with remaining lung function impairment and hypoxemia still warrants further exploring [11–13]. As development of chronic respiratory failure is a feared complication with high morbidity and mortality [14], it is of clinical importance to establish if the overall incidence of chronic hypoxic respiratory failure has changed since the start of the pandemic.

Long-Term Oxygen Therapy (LTOT) is an established treatment to improve survival in patients with chronic severe daytime hypoxemia [15, 16]. The traditional main underlying causes of LTOT have been chronic obstructive pulmonary disease (COPD), idiopathic pulmonary fibrosis, and pulmonary vascular disease [14]. During the period of 2014–2019, the incidence of starting LTOT in Sweden has been stable at about 1,300 patients per year [14]. However, the general impact of the covid-19 pandemic on the prescription and pattern of LTOT on a national level have not been evaluated. The Swedish National Registry of

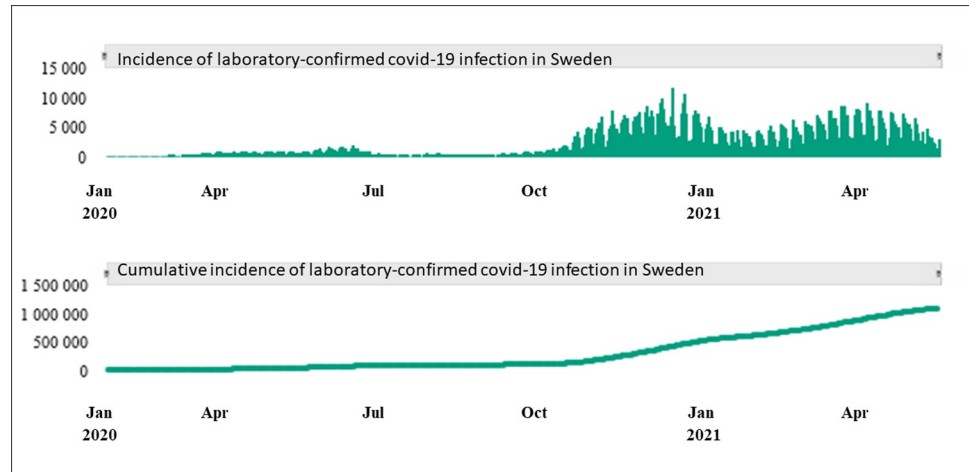

**Fig 1. Incidence of laboratory confirmed covid-19 in Sweden.** Incidence and cumulative incidence of laboratory confirmed covid-19 in Sweden. Data from the Swedish Public Agency [2]. *Abbreviations: Jan = January, Apr = April, Jul = July, Oct = October.*

Respiratory Failure (Swedevox) is a world unique data source on patients with oxygen-dependent chronic respiratory failure, and includes about 85% of all patients starting LTOT since 1987 nationwide [14].

The main aims of this study were to explore the incidence of LTOT due to covid-19 as well as the total incidence of LTOT in Sweden between March and December 2020, and to compare the incidence of LTOT and characteristics of Swedish patients starting LTOT during the same months during 2015–2019, with the hypothesis that the overall incidence of LTOT had increased due to covid-19. We also aimed to investigate whether the working load and tasks of Swedish oxygen nurses have changed during the pandemic.

## Material and methods

### Study subjects and design

This was a retrospective cohort study of patients aged $\geq$ 18 years starting LTOT and reported to the Swedevox registry in Sweden. Patients on palliative oxygen or oxygen used only at exertion was not included.

### Assessments

Data was derived from the Swedevox register on incidence of LTOT for each year 2015–2020, prevalence of LTOT December 2020, and on patient characteristics in terms of sex, age, body mass index (BMI), smoking status, underlying diseases, Eastern Cooperative Oncology Group-World Health Organization (ECOG-WHO) performance status [17], Dyspnoea Exertion Scale (DES) for breathlessness [18], forced expiratory volume in one second ($FEV_1$), vital capacity (VC), baseline hypoxemia ($PaO_2$ and $SpO_2$ breathing air and oxygen) and prescribed flow (l/min) of oxygen. WHO status extends from 0 denoting full activity without restrictions to 4 meaning being completely disabled [17]. DES is a six-point scale assessing breathlessness, from 0 (being able to walk at one's own pace without getting out of breath) to 5 (being breathless at rest) [18].

A web-based questionnaire was emailed to the responsible oxygen nurses at all units prescribing LTOT in Sweden, to assess the impact of covid-19 on LTOT during 2020. The first question was "How many patients has received LTOT during 2020 after or due to covid-19"? The second question was "How has the number of patients starting therapeutic LTOT been affected by the pandemic compared with previous years?" (select the most appropriate alternative; much fewer; slightly fewer; unchanged; slightly more; or much more). The third question was "Have your working tasks changed due to covid-19?" (select the most appropriate alternative; yes, much less or not at all work with oxygen patients; yes, a little less work with oxygen patients; unchanged; or more work with oxygen patients). The final question was "Has the pandemic lead to fewer patients starting LTOT being registered in Swedevox?" (select suitable alternative; yes/no). As covid-19 is not recorded as specific cause of LTOT in Swedevox, local data on LTOT starts at each centre were used to complete the questionnaire.

### Statistical analyses

The overall incidence of LTOT for the entire year of 2020 compared to the previous five years, as well as the prevalence of LTOT in the end of 2020 were calculated based on data from the Swedovox register. The incidence for LTOT was also calculated for the period of January to December each year) of 2014 to 2020. In units responding to the item of estimated number of LTOT starts due to covid-19 in 2020, LTOT-starts due to covid-19 were calculated both as absolute numbers and as proportion of total LTOT starts at respective units during 2020.

For comparison of characteristics of patients starting LTOT before and after the outbreak of the pandemic, two groups were created. The first group was all patients starting LTOT during March to December 2020 when the pandemic was manifest in Sweden, and the comparison group was patients starting LTOT during the same months between 2014 and 2019. Baseline patient characteristics were tabulated using mean (standard deviation [SD]) for normally distributed continuous variables, median (interquartile range [IQR]) for continuous variables with skewed distributions and frequency (percentage) for categorical variables. Differences between groups were analysed using chi-squared tests, Students t-tests and Mann-Whitney U-tests as appropriate. Due to the large number of assessed variables, the Bonferroni equation of $\alpha / n = 0.05$ was used to calculate the $p$-value. As the number of assessed variables was 15, a p-value of 0.003 was considered as statistically significant.

In the subset of patients with registered one-year follow-up data, the proportion of patients with withdrawal of LTOT due to improvement was calculated.

Statistical analyses were conducted using the software packages Stata, version 16.0 (StataCorp LP; College Station, TX, USA) and IBM SPSS version 25 (IBM Corporation, Armonk, NY, USA).

## Ethical considerations

The study was conducted in accordance with the ethical principles of the revised Declaration of Helsinki. In accordance with Swedish legislation and regulations, patients starting LTOT receive oral and written information about Swedevox including the choice to opt-out from registration in Swedevox or for their data to be removed from the register at any time. This procedure with waived informed consent is approved by all regional ethics committees in Sweden, the National Board of Health and Welfare, and the Data Inspection Board. The subsequent analysis of register data on a group level without further consent from the included patients was approved by the Lund University Research Ethics Committee (Dnr: 2010/315 and 2011/722).

## Results

### LTOT incidence and patient characteristics

Fig 2 shows the incidence of LTOT in Sweden during 2020 compared with the previous five years. In 2020, 989 patients started LTOT compared with a mean of 1,113 patients per year during 2015–19 (n = 1,211, 1,128, 1,030, 1,030 and 1,084 per successive year, respectively). These numbers correspond to an incidence of 12, 11, 10, 10 and 10/100 000 inhabitants, respectively, during those years. At the end of 2020, the prevalence of patients with LTOT in Sweden was 21.9 per 100 000 inhabitants. In total, 89% of patients starting LTOT in Sweden during 2020 had a $PaO_2 < 7.4$ kPa and 99% had a degree of hypoxemia ($PaO_2 < 8$ kPa).

The total estimated number of patients starting LTOT during March-Dec 2020 due to covid-19 was 134, corresponding to 20.5% of the total incidence of LTOT starts 2020.

Table 1 shows patient characteristics of patients starting LTOT during March to December 2020 compared with the corresponding ten months for the years 2015 to 2019. Characteristics of patients starting LTOT were similar across the years with no substantial differences in either demographics or underlying main or additional diseases before and after the outbreak of covid-19 in Sweden (Table 1).

In a subgroup pf patients, n = 2559, one-year follow-up data were available. The number of patients with one-year data were lower in the 2020 group than the 2015–2019 group (n = 65 (9%) vs 2494 (56%), p < 0.001. In patients with one-year follow-up data, a statistically

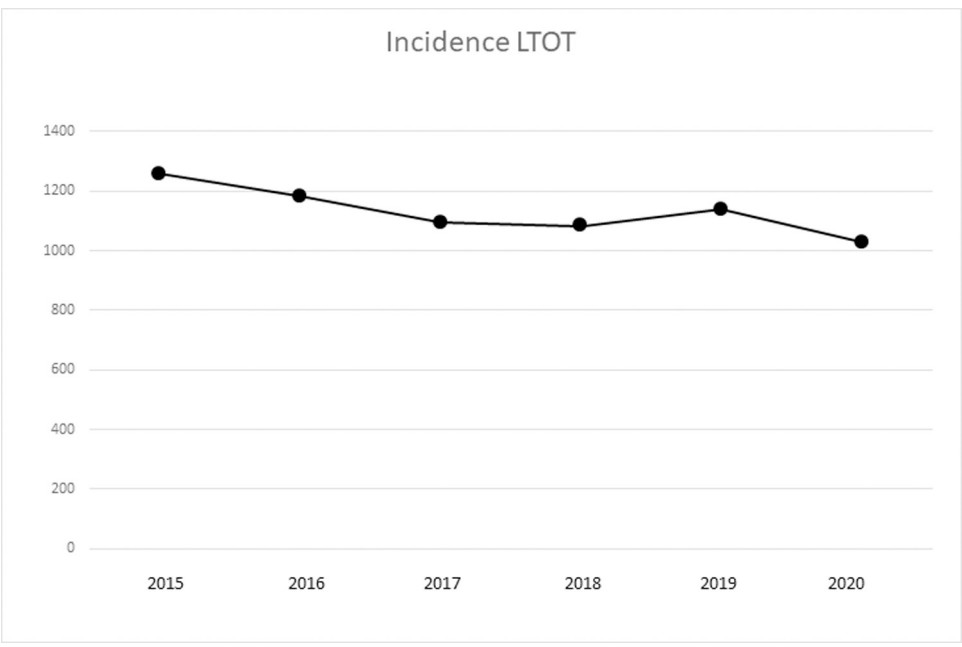

**Fig 2. Incidence of LTOT during 2020 compared with the previous 5 years.** Incidence of LTOT starts in Sweden per whole year 2015 to 2020. *Abbreviations: LTOT = Long Term Oxygen Therapy.*

significantly higher proportion of patients with LTOT start 2020 compared with 2015–2019 were able to withdraw treatment due to improvement (15% vs 7%), p = 0.011.

## Impact on LTOT centres

Representatives from 43 out of 46 LTOT units (93%) responded to at least one of the items in the online questionnaires. The reported mean number of LTOT starts due to covid-19 was 3.7 (SD 5.4). The majority of the centres prescribing LTOT reported that the covid-19 pandemic had not changed the overall incidence of LTOT during 2020 (Fig 3). As for the tasks of oxygen nurses, the two most common responses were that they had unchanged or slightly less time for work with LTOT patients (Fig 3), and 24% reported that there had not been the same time for registering new LTOT patients into Swedevox.

## Discussion

### Main findings

The first main finding was that covid-19 accounted for a substantial part (20.5%) of new treatments with LTOT nationally in 2020. Secondly, the total incidence of LTOT was not increased in relation to the pandemic and characteristics of patients starting the therapy were similar before and after the outbreak of the pandemic. Thirdly, more than half of the LTOT units had to adjust their work due to the pandemic.

This is the first study evaluating the impact of covid-19 on the need for LTOT on a national level. Strengths of this study are that data from Swedevox has a high degree of internal and external validity due to the quality of data, the high geographical coverage and completeness of the register [19]. A limitation is that the questionnaire provided estimated data, as covid-19 was not included as a variable directly in Swedevox, and therefore could be subject to recall bias. However, according to regular assessments, LTOT prescription and management is

**Table 1. Characteristics of patients starting LTOT in 2020 compared with previous five years.**

| LTOT starting date | Mar-Dec, 2020 | Mar-Dec, 2015–19 | p-value |
|---|---|---|---|
| Female sex | 419 (56) | 2597 (58) | 0.25 |
| Age (years) | 75.4 (10.2) | 75.6 (9.4) | 0.67 |
| Smoking status | | | 0.048 |
| Never smoker | 113 (18) | 598 (15) | |
| Ex-smoker | 496 (81) | 3266 (83) | |
| Smoker | 4 (1) | 55 (1) | |
| BMI (kg/m$^2$) | | | 0.043 |
| <20.0 | 94 (16) | 695 (19) | |
| 20.0 to 24.9 | 203 (33) | 1270 (34) | |
| 25.0 to 29.9 | 146 (24) | 952 (25) | |
| ≥30.0 | 165 (27) | 842 (22) | |
| Main diagnosis | | | 0.449 |
| Airway disease | 440 (60) | 2762 (62) | |
| Parenchymal disease | 150 (20) | 864 (19) | |
| Pulmonary vascular disease | 39 (5) | 271 (6) | |
| Heart disease | 38 (5) | 202 (5) | |
| Other | 69 (9) | 352 (8) | |
| Additional diagnosis | | | 0.254 |
| Airway disease | 19 (8) | 69 (6) | |
| Parenchymal disease | 19 (8) | 87 (7) | |
| Pulmonary vascular disease | 35 (15) | 151 (12) | |
| Heart disease | 71 (31) | 439 (36) | |
| Other | 86 (37) | 479 (39) | |
| WHO performance status | | | 0.107 |
| 0 | 45 (8) | 383 (11) | |
| 1 | 268 (49) | 1514 (43) | |
| 2 | 152 (28) | 1035 (30) | |
| 3 | 74 (14) | 504 (14) | |
| 4 | 11 (2) | 66 (2) | |
| DES breathlessness scale | | | 0.981 |
| 1 | 11 (3) | 52 (2) | |
| 2 | 67 (16) | 378 (17) | |
| 3 | 132 (32) | 701 (31) | |
| 4 | 94 (23) | 535 (24) | |
| 5 | 77 (19) | 427 (19) | |
| 6 | 34 (8) | 169 (8) | |
| PaO$_2$ (air) (kPa) | 6.53 (0.81) | 6.47 (0.85) | 0.096 |
| SpO$_2$ (air) (kPa) | 81.0 (7.6) | 81.7 (2.7) | 0.50 |
| PaO$_2$ (oxygen) (kPa) | 8.64 (1.23) | 8.61 (1.69) | 0.65 |
| SpO$_2$ (oxygen) (kPa) | 92.0 (4.38) | 92.8 (2.68) | 0.14 |
| FEV$_1$ (l) | 1.21 (0.63) | 1.18 (0.67) | 0.42 |
| VC (l) | 2.13 (0.80) | 2.13 (0.80) | 0.70 |
| Prescribed O$_2$ flow (l/min) | 1.5 [1.0–2.0] | 1.5 [1.0–2.0] | 0.88 |

Data compared between time periods for the same calendar months (Mar-Dec), presented as mean (standard deviation [SD]) for normally distributed continuous variables, median (interquartile range [IQR] for continuous variables with skewed distributions and frequency (percentage) for categorical variables. Differences between groups were analysed using chi-squared tests, Students t-tests and Mann-Whitney U-tests. Missing data were for smoking status: n = 674, BMI: n = 839, main diagnosis: n = 19, additional diagnosis: n = 3751, WHO performance status: n = 22, DES breathlessness scale: n = 414, PaO$_2$ (air): n = 1089, SpO2 (air): n = 4706, PaO$_2$ (oxygen): n = 778, SpO2 (oxygen): n = 4710, FEV1: n = 2276, VC: n = 2368, and for prescribed oxygen flow: n = 25. *Abbreviations: LTOT = Long Term Oxygen Therapy, Mar = March, Dec = December, BMI = Body Mass Index, WHO = World Health Organization, DES = Dyspnoea Exertion Scale, FEV1 = Forced expiratory volume in one second, VC = Vital Capacity.*

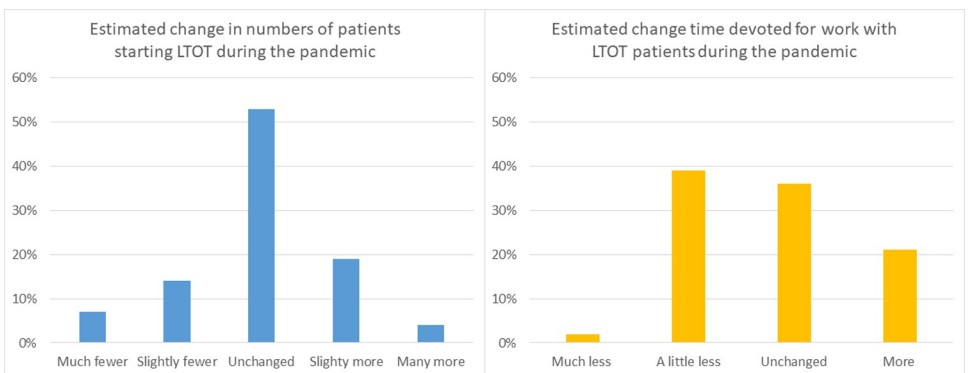

**Fig 3. Change in LTOT incidence and working time with LTOT during the pandemic.** Estimated impact of the covid-19 pandemic on LTOT starts and work in Sweden. *Abbreviations*: *LTOT = Long Term Oxygen Therapy*.

generally well-structured in Sweden [14]. LTOT is prescribed and followed-up by specialists in respiratory units (and is not prescribed in primary care or private outpatient clinics), centres have lists on active patients and the number of patients for the oxygen nurses per centre is generally reasonable to enable oversight over the patients starting LTOT over the preceding year. Even if the exact number of starts due to covid-19 could not be obtained from Swedevox, the questionnaire data on LTOT starts due to covid-19 is based on local recorded information at each specialised site which forms the basis of the given care.

The choice to compare incidence of LTOT from March to December 2020 with previous years could be discussed, as the majority of LTOT starts due to covid-19 did not happen until a couple of months later. However, we choose to compare March to December to include the main period of the covid-19 pandemic in Sweden during 2020. The reason for investigating only 2020 and not 2021 is that the questionnaire about LTOT starts during the pandemic was completed by the LTOT centres in the end of 2020, in order to investigate the impact of the early phase before mass-vaccination of COVID-19 started.

We deliberately chose to include patients prescribed therapeutic LTOT with the aim of prolonging life, to patients fulfilling the BTS recommendations of hypoxemia at rest [20]. In Sweden, the BTS recommendations of indication for LTOT are followed very strictly [14]. This means that both patients with milder sequele after COVID-19 using oxygen at training during rehabilitation, as well as patients with severe end stage COVID-19 as palliation are excluded. However, our aim was to study the need of long-term oxygen in remaining chronic hypoxic respiratory failure. The unchanged total LTOT incidence despite increased starts due to covid-19 may have different reasons. Exacerbations with infections strongly contribute to declining lung function and development of respiratory failure in patients with COPD and other lung disease, but the recommended behaviour of avoiding social contacts and improvement in hygiene routines during the pandemic may have decreased infections and hence development of incident respiratory failure in this patient group. This speculation is supported by a review showing generally decreased number of hospitalisations due to COPD exacerbations after the start of the pandemic, all over the world [21]. Specifically, this has been reported in studies from Norway and Denmark [22, 23]. Norway, Denmark and Sweden are Nordic neighbour countries and the similarity between the countries makes it reasonable to expect the same pattern in Sweden. In addition, we cannot exclude the possibility of a displacement effect from the pandemic, partly because of increased waiting times for revisits for patients with chronic lung diseases.

We speculate that the excess mortality of covid-19 may have affected patients with existing respiratory diseases with incipient respiratory failure, and thus explain why the total incidence

of LTOT have not increased. Potential explanations to why the proportion of underlying air-way disease in relation to parenchymal disease has not decreased may be that many patients who started LTOT due to covid-19 had underlying airway diseases that would have caused respiratory failure during this period even without the suffered covid-19 infection, and that the covid-19 infection has not been reported as underlying cause in those cases. This hypothesis could be supported by the high number of missing data for additional but not for main under-lying diseases.

Interestingly, data from the subset of patients with follow-up data showed that the propor-tion of withdrawal of LTOT due to improvement was more common during the study period during 2020. This is in agreement with the fact that covid-19 has been estimated as underlying cause in a fifth of LTOT starts during March to December 2020, as several studies has reported clinical, radiological and functional improvement over time in some patients with covid-19 infection [11–13].

The present findings have several potential clinical implications. First, covid-19 has arisen as an important cause of developing chronic respiratory failure in need of LTOT, mainly in elderly patients with underlying diseases. Secondly, the increased need of LTOT caused by covid-19 but not increased total LTOT incidence may, at least partly, be explained by reduced risk of exacerbations due to other respiratory infections. Finally, Swedish LTOT units seem to have adapted to and managed the consequences of the pandemic well.

Interesting research questions that need to be addressed by future studies are if younger and previously healthy patients starting LTOT due to covid-19 may have a more temporary need of LTOT, and if the pattern of underlying diseases in LTOT starts will change during the coming years. It so also of great interest to explore if the increased incidence of covid-19 dur-ing 2021 will lead to an overall increase in need of LTOT, or be counteracted by vaccination strategies and new treatments. Future studies should also proceed from the population with confirmed covid-19-infection linked to national registers, and investigate the true incidence of LTOT in covid-19 compared with other causes.

In conclusion, the incidence of LTOT in Sweden has not increased after the outbreak of the pandemic. In spite of a clear number of patients with covid-19 starting LTOT during 2020, the total incidence of LTOT and the characteristics of patients starting LTOT are the same before and after the outbreak of covid-19. We speculate that the increase in LTOT due to covid-19 is outweighed by a reduction in worsening because of other infections avoidance of regular visits and an excess mortality before being discharged and evaluated for LTOT.

## Acknowledgments

We thank all staff at the participating centres for reporting to Swedevox and caring for the patients.

## Author Contributions

**Conceptualization:** Magnus Ekström.

**Formal analysis:** Josefin Sundh.

**Methodology:** Josefin Sundh, Andreas Palm, Josefin Wahlberg, Michael Runold, Magnus Ekström.

**Writing – original draft:** Josefin Sundh.

**Writing – review & editing:** Andreas Palm, Josefin Wahlberg, Michael Runold, Magnus Ekström.

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
