## [Decision Letter · Decision Letter 0]

7 Feb 2022

PONE-D-21-37749Impact of covid-19 on long-term oxygen therapy 2020: a nationwide study in SwedenPLOS ONE

Dear Dr. Josefin Sundh,

Thank you for submitting your manuscript to PLOS ONE. After careful consideration, we feel that it has merit but does not fully meet PLOS ONE’s publication criteria as it currently stands. Therefore, we invite you to submit a revised version of the manuscript that addresses the points raised during the review process.

This study reports a lower incidence of LTOT in Sweden between March and December 2020, in comparison to the incidence of LTOT in the same months during 2015-2019. The main hypothesis is that the overall incidence of LTOT had increased due to covid-19. Authors estimated that Covid-19 caused one fifth of all new LTOT during the pandemic in 2020. Despite having a solid source of data represented by the National Registry of Respiratory Failure, reviewers arise major concers, which need to be better discussed and explored.Moreover, I questioned if patients with respiratory comorbidity (mainly with COPD) that died from COVID-19 and the reduction of follow-up visits expecially during the first wave could represent another reason of lower new OTLT during 2020.   Please submit your revised manuscript by Mar 24 2022 11:59PM. If you will need more time than this to complete your revisions, please reply to this message or contact the journal office at plosone@plos.org. Please include the following items when submitting your revised manuscript:

We look forward to receiving your revised manuscript.

Kind regards,

Manlio Milanese

Academic Editor

PLOS ONE

Journal Requirements:

(ME was supported by unrestricted grants from the Swedish Society for Medical Research and the Swedish Research Council (Dnr 2019-02081).)

Additional Editor Comments:

This study reports a lower incidence of LTOT in Sweden between March and December 2020, in comparison to the incidence of LTOT in the same months during 2015-2019. The main hypothesis is that the overall incidence of LTOT had increased due to covid-19. Authors estimated that Covid-19 caused one fifth of all new LTOT during the pandemic in 2020. Despite having a solid source of data represented by the National Registry of Respiratory Failure, reviewers arise major concers, which need to be better discussed and explored. Moreover, I questioned if patients with respiratory comorbidity (mainly with COPD) that died from COVID-19 could represent another reason of lower new OTLT during 2020.

Reviewers' comments:

Reviewer's Responses to Questions

**Comments to the Author**

1. Is the manuscript technically sound, and do the data support the conclusions?

Reviewer #1: Partly

Reviewer #2: Yes

2. Has the statistical analysis been performed appropriately and rigorously? 

Reviewer #1: Yes

Reviewer #2: Yes

3. Have the authors made all data underlying the findings in their manuscript fully available?

Reviewer #1: Yes

Reviewer #2: Yes

4. Is the manuscript presented in an intelligible fashion and written in standard English?

Reviewer #1: Yes

Reviewer #2: Yes

5. Review Comments to the Author

Reviewer #1: The authors present an interesting analysis of LTOT dispensings in Sweden in 2020. The 2020 numbers were lower than those seen in the previous 5 years.

Major comments:

While I believe this paper accurately describes LTOT dispensings in Sweden in 2020- I also think that it may not fully be capturing the affect of of COVID. In Figure 1, COVID cases did not surge in Sweden until late in 2020 ( and most of 2021). Thus- I suspect this same study extended into 2021 might look different.

Are data available on a metric such as COPD hospitalizations over this time period? The US saw a drop in these during 2020 ( probably related to social distancing). I suspect this may have also been a factor in places like Sweden.

Reviewer #2: General comment

The main aims of this study were to explore the incidence of LTOT due to covid-19 as well as the total incidence of LTOT in Sweden between March and December 2020, in comparison to the incidence of LTOT in the same months during 2015-2019. The main hypothesis is that the overall incidence of LTOT had increased due to covid-19. The major results were that the total number of patients starting LTOT did not increase and no significant differences in patient characteristics or underlying causes of hypoxemia were found between patients starting LTOT during 2020 compared to 2015-2019. Authors estimated that Covid-19 caused one fifth of all LTOT starts during the pandemic in 2020. Despite having a solid source of data represented by the National Registry of Respiratory Failure, the paper has good purposes, but has some weak points which need to be better discussed and explored.

Major corrections

1) Introduction: After two years of the pandemic, numerous studies have concluded that during the follow-up Lung volumes, DLCO, chest radiographic abnormalities and respiratory symptoms tend to normalize or improve 1 year after hospitalization for COVID-19 in most patients. Thus, it is difficult to hypothesize that Covid-19 is directly responsible for chronic respiratory failure (Truffaut et al. Respir Res 2021; 22:29 https://doi.org/10.1186/s12931-021-01625-y) (Balbi M et al. Eur J Radiol 2021; 138 https://doi.org/10.1016/j.ejrad.2021.109676). Please, in the light of these findings, define better the background and aim of the study in the background.

2) The choice of the reference period is questionable. In the six months between October 2020 and March 2021, the Sweden saw 657 ,309 positive cases in comparison to just under 93 ,000 cases by 1 October 2020 since the onset of pandemic. (Paterlini M, BMJ 2021; 375 doi: https://doi.org/10.1136/bmj.n3081). The authors described in Figure 1 the higher incidence of cases in the second and third pandemic waves. It is reasonable to believe, as partly discussed by the authors, that the initiation of LTOT is mainly attributable to underlying pre-existing chronic disease, since Covid-19 may have a more temporary need of oxygen therapy. It is not clear, in fact, the prescribing criteria of LTOT in Covid-19 patients, considering that the estimate of patients with Covid-19 is obtained only from local records. Do Authors think that an extension of the enrollment period could provide more robust evidence regarding hypothetical prescribing changes related to the pandemic?

3) Methods: ‘Patients on palliative oxygen or oxygen used only at exertion was not included’. Do authors think that this choice may have excluded those patients suffering from sequelae of severe forms of Covid-19 related pneumonia (i.e. DNR patients or patients requiring a rehabilitation period)?

4) It would be interesting to have more detailed clinical data on Covid-19 patients who started LTOT. Were patients with severe pneumonia or sequelae treated in ICU? How many patients were hospitalized? Did patients have any concomitant chronic diseases? Did any patient eventually discontinue the LTOT after a rehabilitation period or 1-year post-discharge?

5) Do you think it is possible to identify patients who have required prolonged weaning from ventilation or tracheostomy for Covid-19 pneumonia? This subgroup of hospitalized patients may have been prescribed by OTLT. Expecting a higher hospitalization rate during the second and third Covid-19 waves, it is reasonable to assume a higher number of patients eligible for OTLT by extending the referral / enrollment period.

Minor corrections

1) Please, indicate the estimate updated date of the sentence ‘Up to date, the global number of people diagnosed with covid-19 is over 180 million, and almost 4 million have died due to the disease’.

2) Please specify units of measurement in table and graphs.

3) Please verify in the text %, use the same style: i.e. Number and ‘%’ without space character (i.e. 20%)

4) Please verify in the text, use the same style for thousands separator, authors used ‘,’ and space characters (i.e. 234 665 or 234,665)

6. PLOS authors have the option to publish the peer review history of their article (what does this mean?). If published, this will include your full peer review and any attached files.

Reviewer #1: **Yes: **David Mannino

Reviewer #2: No

---

## [Author Response · Author response to Decision Letter 0]

2 Mar 2022

Dear Editor, 

Thank you for the valuable review comments relating to our manuscript “PONE-D-21-37749- Impact of covid-19 on long-term oxygen therapy 2020: a nationwide study in Sweden”. 

We have carefully read and taken the concerns into consideration. Our responses to the editors and reviewers’ comments are listed below. A revised version of our manuscript with tracked changes has been uploaded, as well as a clean revised version,

Thank you again for giving us this chance to improve our manuscript. 

Yours sincerely

Josefin Sundh

Journal Requirements:

Response: We have adjusted the headings and structure of the manuscript according to the PLOS ONE template.

(ME was supported by unrestricted grants from the Swedish Society for Medical Research and the Swedish Research Council (Dnr 2019-02081).)

Response: We understand and have added the sentence “There was no additional external funding received for this study” into the manuscript and in a separate new cover letter.

Response: According to the Swedish law, data cannot be made freely available as they are subject to secrecy in accordance with the Swedish Public Access to Information and Secrecy Act, but can be made available to researchers upon request. We have previously specified this in a Data sharing statement in the main file, and have now added information that Data sharing needs approval from the Swedish Ethical Review Authority. We have also added the Data sharing statement in a separate new cover letter.

Additional Editor Comments:

This study reports a lower incidence of LTOT in Sweden between March and December 2020, in comparison to the incidence of LTOT in the same months during 2015-2019. The main hypothesis is that the overall incidence of LTOT had increased due to covid-19. Authors estimated that Covid-19 caused one fifth of all new LTOT during the pandemic in 2020. Despite having a solid source of data represented by the National Registry of Respiratory Failure, reviewers arise major concers, which need to be better discussed and explored. Moreover, I questioned if patients with respiratory comorbidity (mainly with COPD) that died from COVID-19 could represent another reason of lower new OTLT during 2020.

Response: We will carefully consider the review comments below, and follow their recommendations.

We fully agree about the possibility that patients with respiratory comorbidity at risk for LTOT may have died and thereby lowered the incidence of LTOT. This speculation is already mentioned in page 11 lines 265-267(tracked revised version).

Reviewer #1: The authors present an interesting analysis of LTOT dispensings in Sweden in 2020. The 2020 numbers were lower than those seen in the previous 5 years.

Major comments:

While I believe this paper accurately describes LTOT dispensings in Sweden in 2020- I also think that it may not fully be capturing the affect of COVID. In Figure 1, COVID cases did not surge in Sweden until late in 2020 (and most of 2021). Thus- I suspect this same study extended into 2021 might look different.

Are data available on a metric such as COPD hospitalizations over this time period? The US saw a drop in these during 2020 ( probably related to social distancing). I suspect this may have also been a factor in places like Sweden.

Response: We fully agree that the impact of COVID-19 may have changed during 2021, but our aim with this study was to describe the overall impact on LTOT prescription during the first year of the pandemic. Unfortunately we have no data on overall frequency of exacerbations or hospitalizations due to COPD in Sweden before and after the start of the pandemic, but we have the same opinion that the social distancing and subsequent exacerbations are likely to explain why the incidence of LTOT did not increase in spite of the COVID-19 pandemic. We have discussed this in page 12 lines 252 to 256, and have extended the discussion in page 12 line 257 to page 13 line 262 including references to a review describing the generally decreased hospitalizations due to COPD exacerbations in the world, and to specific studies from our neighbor countries Norway and Denmark which support the speculation. 

Reviewer #2: General comment

The main aims of this study were to explore the incidence of LTOT due to covid-19 as well as the total incidence of LTOT in Sweden between March and December 2020, in comparison to the incidence of LTOT in the same months during 2015-2019. The main hypothesis is that the overall incidence of LTOT had increased due to covid-19. The major results were that the total number of patients starting LTOT did not increase and no significant differences in patient characteristics or underlying causes of hypoxemia were found between patients starting LTOT during 2020 compared to 2015-2019. Authors estimated that Covid-19 caused one fifth of all LTOT starts during the pandemic in 2020. Despite having a solid source of data represented by the National Registry of Respiratory Failure, the paper has good purposes, but has some weak points which need to be better discussed and explored.

Major corrections

1) Introduction: After two years of the pandemic, numerous studies have concluded that during the follow-up Lung volumes, DLCO, chest radiographic abnormalities and respiratory symptoms tend to normalize or improve 1 year after hospitalization for COVID-19 in most patients. Thus, it is difficult to hypothesize that Covid-19 is directly responsible for chronic respiratory failure (Truffaut et al. Respir Res 2021; 22:29 https://doi.org/10.1186/s12931-021-01625-y) (Balbi M et al. Eur J Radiol 2021; 138 https://doi.org/10.1016/j.ejrad.2021.109676). Please, in the light of these findings, define better the background and aim of the study in the background.

Response: We agree that there is plenty of evidence that both radiological impairment and DLCO after COVID usually improves over time, and have added the suggested references in the introduction. However, the fact that a minority of patients still may have remaining lung function impairment and hypoxemia is an important rationale for the present study, as the pandemic affect such a large number of people across the population, and as chronic respiratory failure is associated with very high morbidity and mortality. We have added a paragraph in the introduction discussing this issue (page 3 line 72 to page 4 line 77)

2) The choice of the reference period is questionable. In the six months between October 2020 and March 2021, the Sweden saw 657,309 positive cases in comparison to just under 93 ,000 cases by 1 October 2020 since the onset of pandemic. (Paterlini M, BMJ 2021; 375 doi: https://doi.org/10.1136/bmj.n3081). The authors described in Figure 1 the higher incidence of cases in the second and third pandemic waves. It is reasonable to believe, as partly discussed by the authors, that the initiation of LTOT is mainly attributable to underlying pre-existing chronic disease, since Covid-19 may have a more temporary need of oxygen therapy. It is not clear, in fact, the prescribing criteria of LTOT in Covid-19 patients, considering that the estimate of patients with Covid-19 is obtained only from local records. Do Authors think that an extension of the enrollment period could provide more robust evidence regarding hypothetical prescribing changes related to the pandemic?

Response: We acknowledge the fact that the number of COVID-19 cases was much lower during the study period until Dec 2020 than during the year after. However, our intention in this study was to compare LTOT need during 2020 when the pandemic first occurred, with previous years. We find it important to study the early phase before mass vaccination started, but we fully agree that future follow-up studies with extended enrollment period are needed too. We have tried to clarify this choice in page 12 lines 243-244.

3) Methods: ‘Patients on palliative oxygen or oxygen used only at exertion was not included’. Do authors think that this choice may have excluded those patients suffering from sequelae of severe forms of Covid-19 related pneumonia (i.e. DNR patients or patients requiring a rehabilitation period)?

Response: Our intention was to study the impact of the pandemic on numbers of patients developing chronic hypoxemia in need of LTOT according to the BTS recommendations. We agree that this means that patients with only effort hypoxemia after COVID-19 are excluded. However, as the criteria for LTOT are based on hypoxemia at rest, studying hypoxemia at effort was not the purpose of this study. 

As for palliative oxygen treatment, this refers to oxygen not prescribed with the purpose of prolonging life, but rather as symptomatic treatment in end stage disease. We fully agree that we must have excluded a considerable number of patients receiving oxygen due to severe end-stage COVID-19 with hypoxemia, but due to their short survival these patients are not representative of the population with long-term sequelae of COVID-19 with need of LTOT. We have added a paragraph discussing these issues in page 12 lines 245 to 251.

4) It would be interesting to have more detailed clinical data on Covid-19 patients who started LTOT. Were patients with severe pneumonia or sequelae treated in ICU? How many patients were hospitalized? Did patients have any concomitant chronic diseases? Did any patient eventually discontinue the LTOT after a rehabilitation period or 1-year post-discharge?

Response: We agree with the reviewer that more descriptive data would have been of interest, but unfortunately Swedevox do not include information on number of hospitalizations or specific comorbid conditions. All available clinical data are reported in table 1, such as main and additional underlying respiratory conditions, performance status and PaO2 on air. 

One year-follow up data was available only in a minority of the 2020 group; 65 (9%) vs 2494 (56%) in the 2015-2109 group. However, among those with follow-up data after one year, a statistically significantly higher proportion of patients with LTOT start 2020 compared with 2015-2019) were able to withdraw treatment due to improvement (10 (15%) vs 177 (7%). We agree that this is interesting data and have added the analysis in the method (page 7 lines 146 to 147), result (page 10 lines 200 to 204) and discussion sections (page 13 lines 274 to 279).

5) Do you think it is possible to identify patients who have required prolonged weaning from ventilation or tracheostomy for Covid-19 pneumonia? This subgroup of hospitalized patients may have been prescribed by OTLT. Expecting a higher hospitalization rate during the second and third Covid-19 waves, it is reasonable to assume a higher number of patients eligible for OTLT by extending the referral / enrollment period.

Response: Unfortunately, we do not have specific data on prolonged weaning from ventilation or tracheostomy in this population. As for the second and third covid-19 waves, we fully agree that it is of high interest with future follow-studies to study if the impact on LTOT starts will change. Future need of studies is discussed in page 14 lines 286 to 293. 

Minor corrections

1) Please, indicate the estimate updated date of the sentence ‘Up to date, the global number of people diagnosed with covid-19 is over 180 million, and almost 4 million have died due to the disease’.

Response: The original manuscript was written in June 2021, and the numbers in the introduction were from May 2021. However, we agree that it is reasonable to update the general global numbers and have added the information in the introduction section.

2) Please specify units of measurement in table and graphs.

Response: We have followed the advice and added units in the table.

3) Please verify in the text %, use the same style: i.e. Number and ‘%’ without space character (i.e. 20%)

Response: We have followed the advice and changed the typos.

4) Please verify in the text, use the same style for thousands separator, authors used ‘,’ and space characters (i.e. 234 665 or 234,665)

Response: We have followed the advice and changed to correct separations with “,”.

---

## [Decision Letter · Decision Letter 1]

21 Mar 2022

Impact of covid-19 on long-term oxygen therapy 2020: a nationwide study in Sweden

PONE-D-21-37749R1

Dear Dr. Josefin Sundh,

We’re pleased to inform you that your manuscript has been judged scientifically suitable for publication and will be formally accepted for publication once it meets all outstanding technical requirements.

Kind regards,

Manlio Milanese

Academic Editor

PLOS ONE

Additional Editor Comments (optional):

Comment to Reviewers were adequately addressed. The manuscript is now imporved and suitable for publication on PlosOne.

Reviewers' comments:

Reviewer's Responses to Questions

**Comments to the Author**

1. If the authors have adequately addressed your comments raised in a previous round of review and you feel that this manuscript is now acceptable for publication, you may indicate that here to bypass the “Comments to the Author” section, enter your conflict of interest statement in the “Confidential to Editor” section, and submit your "Accept" recommendation.

Reviewer #1: All comments have been addressed

Reviewer #2: All comments have been addressed

2. Is the manuscript technically sound, and do the data support the conclusions?

Reviewer #1: Yes

Reviewer #2: Yes

3. Has the statistical analysis been performed appropriately and rigorously? 

Reviewer #1: Yes

Reviewer #2: Yes

4. Have the authors made all data underlying the findings in their manuscript fully available?

Reviewer #1: Yes

Reviewer #2: Yes

5. Is the manuscript presented in an intelligible fashion and written in standard English?

Reviewer #1: Yes

Reviewer #2: Yes

6. Review Comments to the Author

Reviewer #1: My prior comments have been addressed. I would like to see this replicated when 2021 data becomes available- as the findings may differ!

Reviewer #2: The manuscript contains adequate insights and the authors have answered exhaustively to the questions and requests for revision.

Minor revision

Please provide to use in the paper the same acronym for Covid-19

Line 75: correct it into It

Line 168: correct 10/100 000 into 10/100,000

Line 170: correct 100 000 into 100.000

Please verify in the paper, tables and footnotes to write thousands numbers in the same style

Line 202: correct one.year into one-year

Line 245: please consider to write in full and cite BTS guidelines in references

7. PLOS authors have the option to publish the peer review history of their article (what does this mean?). If published, this will include your full peer review and any attached files.

Reviewer #1: **Yes: **David Mannino

Reviewer #2: No

---

## [Editor Report · Acceptance letter]

31 Mar 2022

PONE-D-21-37749R1 

Impact of covid-19 on long-term oxygen therapy 2020: a nationwide study in Sweden 

Dear Dr. Sundh:

I'm pleased to inform you that your manuscript has been deemed suitable for publication in PLOS ONE. Congratulations! Your manuscript is now with our production department. 

Kind regards, 

on behalf of

Dr. Manlio Milanese 

Academic Editor

PLOS ONE